# DIFFERENTIABLE SELF-ADAPTIVE LEARNING RATE

## ABSTRACT

Adaptive learning rate has been studied for a long time. In the training session of neural networks, learning rate controls update stride and direction in a multi-dimensional space. A large learning rate may cause failure to converge, while a small learning rate will make the convergence too slow. Even though some optimizers make learning rate adaptive to the training, e.g., using first-order and second-order momentum to adapt learning rate, their network's parameters are still unstable during training and converges too slowly in many occasions. To solve this problem, we propose a novel optimizer which makes learning rate differentiable with the goal of minimizing loss function and thereby realize an optimizer with truly self-adaptive learning rate. We conducted extensive experiments on multiple network models compared with various benchmark optimizers. It is shown that our optimizer achieves fast and high qualified convergence in extremely short epochs, which is far more faster than those state-of-art optimizers.

## 1 INTRODUCTION

Learning rate is one of the most important hyper-parameters in artificial neural networks, and it is the core of an optimizer[1]. In a training session of neural network, an optimizer takes the duty to update network parameters, and directly affects the training speed and final effect.

The early SGD(stochastic gradient descent) method (Lemaréchal, 2012; Courant et al., 1994; Curry, 1944) uses a fixed learning rate in the training session. Usually, we choose a large learning rate to accelerate training. However, this may cause network's parameter to fluctuate around the extreme point and even cannot converge under SGD (Ruder, 2016), which is a very unstable optimization. Momentum (Sutskever et al., 2013b) takes the exponential moving average of historical gradient as a stride. When parameters get into an oscillation, positive and negative gradients will neutralize with each other, thereby reducing the update stride and getting rid of the fluctuation.

However, learning rate under Momentum cannot adapt to the training session. Actually, learning rate needs to be larger in the beginning of training session and smaller at the later stages. AdaGrad (Duchi et al., 2011) multiply learning with the reciprocal of two-norm of historical gradients to have learning rate reduce along with training session. While AdaGrad is interfered a lot by historical gradient, i.e., learning rate in AdaGrad keeps decreasing along with training resulting that learning rate is too small in the late stage of training. RMSProp (Graves, 2013) replaces two-norm in AdaGrad with exponential moving average of historical gradients to reduce the interference. Adam (Kingma & Ba, 2014) combines Momentum and RMSProp, so that it can solve fluctuations and make learning rate adaptive in the same time. However, Adam still has many limitations. Firstly, Adam usually cannot converge well in the late stages of the training. Secondly, after an oscillation, learning rate is usually very small resulting that it converges very slowly in next several epochs. Thirdly, the update stride is limited by the learning rate. However, learning rate cannot grow quickly when a large stride is needed. In a word, Adam is not sensitive enough.

Therefore, none of the current optimizers are stable and sensitive enough to ensure a fast and high qualified convergence. To achieve these two goals, we propose a novel optimizer DSA(differentiable self-adaptive learning rate), which makes learning rate adaptive quickly and accurately. The learning rate in DSA is differentiable with the goal of minimizing loss function. DSA is able to get rid of oscillation in only one or two epochs, since the learning rate in DSA can increases or decreases very sensitively and significantly with the help of learning rate's gradient.

---

[1]https://en.wikipedia.org/wiki/Learning_rate

Based on above discussions and our researches, contributions of DSA are summarized as follows.

- We are the first to make learning rate in neural network differentiable with the goal of minimizing loss function such that learning rate can adapt with a clear instruction.
- DSA can optimize a wide range of network models and can solve two classical and common problems in machine learning, which are grad loss and hard convergence in the later of a training.
- We conduct extensive experiments on multiple neural network models compared with various state-of-art optimizers. Experimental results have demonstrated the absolute advantage of DSA in speed, stability and performance.

In the remaining of this paper, Section 2 describes the proposed method. Experiments are conducted in Section 3. We overview related work in Section 4. Section 5 draws the conclusions.

## 2 METHOD

In this section, we propose our approach, DSA(differentiable self-adaptive learning rate). In Section 2.1, we declare the motivation of DSA. The algorithm of DSA will be given out in Section 2.2 in detail. Then we will explain how to apply DSA to a training process in Section 2.3

### 2.1 MOTIVATION

In this section, we discuss the motivation of the proposed approach in detail.

The early optimizer SGD updates parameters as formula 1, and has three major defects. Firstly, the fixed learning rate in SGD cannot adapt along with the training session. Therefore, it is not sensitive enough and usually stops convergence in the late stage of training. Secondly, it is slow. $g_t$ is always a small number, so the stride is always small too. As a result, the network converges slowly. Thirdly, it is unstable, which we call it an oscillation as shown in Figure 6. A slightly larger learning rate may cause parameter crosses extreme point repeatedly.

$$\theta = \theta - \eta * g_t \tag{1}$$

Many optimizers have been proposed to solve these problems. Among them, Adam is one of the most stable and efficient as formula 2. In the formula, $\theta$ is network's parameter, and $g_t$ is its gradient in $t$-th step. $m_t$ represents the first-order momentum of gradient, and $v_t$ is the second-order momentum. $\eta$ is the learning rate, and $\epsilon$ is an infinitesimal to avoid being divided by zero.

$$\begin{aligned} m_t &= \beta_1 * m_{t-1} + (1 - \beta_1) * g_t \\ v_t &= \beta_2 * v_{t-1} + (1 - \beta_2) * g_t^2 \\ \theta &= \theta - \eta * \frac{m_t}{\sqrt{v_t} + \epsilon} \end{aligned} \tag{2}$$

Adam substitutes $m_t$ for $g_t$ to make the parameter of neural network more stable along with the training session. $v_t$ amplifies the update stride reasonably and makes learning rate adaptive in each epoch. However, unstable occasion still exist commonly especially when $\theta > \eta$, $\theta$ will get into a long-term oscillation as Figure 6. Additionally, slow convergence always exists after a big oscillation of parameter, because $m_t \approx 0$ at that time while $v_t$ is still large. Apart from that, updating stride always steps around learning rate $\eta$ in the late stage of training and results in failure for further optimization. In a word, current optimizers are still not sensitive and stable enough.

To make the learning rate significantly sensitive enough and supply more stable optimization, we propose DSA(differentiable self-adaptive learning rate). In DSA, we design a loss function $\tilde{L}(\eta)$ with learning rate $\eta$ as its independent variable. $\eta$ is differentiable with the goal of $\min \tilde{L}(\eta)$.Therefore, we can use the gradient of learning rate to update the learning rate to make it more sensitive. We will give the details of DSA in section 2.2.

### 2.2 DSA ALGORITHM

In this section, we firstly introduce the basic idea of DSA and then perform three progressive optimizations to basic DSA.

### 2.2.1 BASIC ALGORITHM

Our target is to make learning rate differentiable to get more sensitive and stable optimizer. To achieve this, we design a two-step training and we will talk about it in the following.

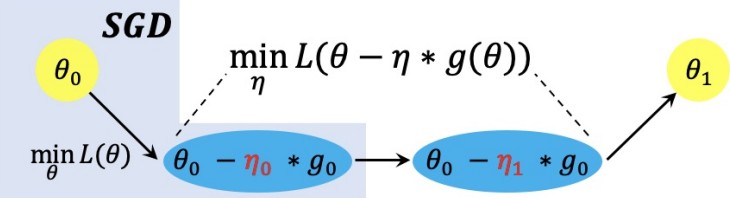

Figure 1: A decomposition diagram of the optimization process within Epoch

In DSA, the training of each epoch consists of two steps as shown in Figure 1. The first step is performed with the goal of $\min_\theta L(\theta)$. The second step is performed with the goal of $\min_\eta \tilde{L}(\eta), \tilde{L}(\eta) \triangleq L(\theta - \eta * g(\theta))$. After the first step, we get a temporary variable $\tilde{\theta}, \tilde{\theta} \triangleq \theta - \eta * g(\theta)$. Actually, $\tilde{\theta}$ is a function with learning rate $\eta$ as its only variable. Therefore, we have the second step rather than assigning $\theta_0 - \eta * g_0$ to $\theta_1$ immediately to take further optimization.

The second step is the core of DSA. Through this step, we achieve two goals. Firstly, the learning rate is differentiable, because $\eta$ is a variable of $\tilde{L}(\eta)$. It means that learning rate can be adaptive to minimize the loss function directly. Therefore, the learning rate will be extremely sensitive. Secondly, the loss function can be optimized further. After the second step, we have $L(\theta_1 - \eta_1 * g_1) < L(\theta_0 - \eta_0 * g_0)$. Actually, $L(\theta_0 - \eta_0 * g_0)$ is the result of SGD, so further optimization is achieved.

When using gradient descent to solve the second step, we will get learning rate's gradient $\sum_{\theta,\tilde{\theta}}(-g(\tilde{\theta}) * g(\theta))$ through derivation in Appendix A. Then iterative equations of DSA can be summarized as formula 3. $g(\theta)$ represents gradient of $\theta$ with the goal of $\min_\theta L(\theta)$. $\beta$ controls the update stride of learning rate $\eta$. In the formula, $\tilde{\theta} = \theta - \eta * g(\theta)$ corresponds to the goal of $\min_\theta L(\theta)$ and $\eta = \eta - \beta * \sum_{\theta,\tilde{\theta}}(-g(\tilde{\theta}) * g(\theta))$ corresponds to the goal of $\min_\eta L(\theta - \eta * g(\theta))$. And finally, parameter $\theta$ is updated according to $\theta = \theta - \eta * g(\theta)$.

$$\tilde{\theta} = \theta - \eta * g(\theta)$$
$$\eta = \eta - \beta * \sum_{\theta,\tilde{\theta}}(-g(\tilde{\theta}) * g(\theta)), \beta > 0 \qquad (3)$$
$$\theta = \theta - \eta * g(\theta)$$

In basic DSA, we use a uniform learning rate for each network's parameter. Actually, these parameters usually have different requirements for learning rate, because they often need to be updated at different speeds. Therefore, we make learning rate parameter specific in the following Section 2.2.2.

### 2.2.2 PARAMETER SPECIFIC LEARNING RATE

In this section, we solve parameter's individual needs for learning rate. To achieve it, we assign a specific learning rate for each parameter. In the following, we first declare the change to iterative equations of DSA, then visualize and explain the process of adapting learning rate in DSA in detail.

If learning rate is specific for each network's parameter, then the gradient of parameter $\theta$ should be changed from $\sum_{\theta,\tilde{\theta}}(-g(\tilde{\theta}) * g(\theta))$ to $-g(\tilde{\theta}) * g(\theta)$. Accordingly, iterative equations of DSA should be changed to formula 4. According to formula 4, if $g(\tilde{\theta}) * g(\theta_{t-1}) > 0$, the learning rate $\eta$ should increase. Otherwise, $\eta$ should decrease.

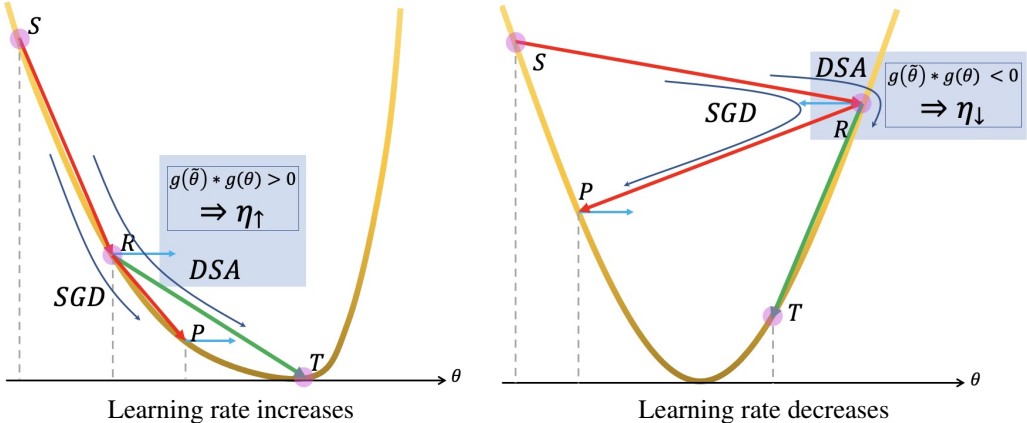

Figure 2: Visualization of learning rate adaptation. Parameter arrives at point $R$ from point $S$ after last iteration and will arrive at point $P$ if we use SGD. S-R-P is track of SGD and S-R-T is track of DSA. $g(\tilde{\theta})$ and $g(\theta)$ corresponds to the gradient of parameter at point $P$ and $R$, respectively.

$$
\begin{aligned}
\tilde{\theta} &= \theta_{t-1} - \eta_{t-1} * g(\theta_{t-1}) \\
\eta_t &= \eta_{t-1} - \beta * (-g(\tilde{\theta}) * g(\theta_{t-1})), \beta > 0 \\
&= \eta_{t-1} + \beta * (g(\tilde{\theta}) * g(\theta_{t-1})) \\
\theta_t &= \theta_{t-1} - \eta_t * g(\theta_{t-1})
\end{aligned}
\tag{4}
$$

Now we use Figure 2 to explain how the learning rate is adaptive in DSA based on formula 4. In the left example, $P$ still haven't reached the extreme point, so learning rate should increase at $R$. In this case, formula 4 also indicates that learning rate should increase because $g(\tilde{\theta}) * g(\theta) > 0$. In the right example, $P$ has crossed the extreme point, so learning rate should decrease at $R$ to avoid that. In this case, formula 4 also indicates that learning rate should decrease because $g(\tilde{\theta}) * g(\theta) < 0$. From both of the two examples, we can conclude that learning rate in DSA is adaptive reasonably.

Even though the learning rate is now parameter specific, DSA can still not deal with the grad loss because $\eta_t * g(\theta_{t-1})$ may be very small sometimes. And we will solve it in Section 2.2.3.

### 2.2.3 EXPONENTIAL LEARNING RATE

In this section, we try to solve grad loss in DSA. We first state the change for DSA to solve the grad loss and then explain why it is reasonable.

Totally, we make two changes to DSA. Firstly, we use exponential the learning rate $2^\eta$ instead of $\eta$. Secondly, we use $\frac{g(\theta)}{|g(\theta)|+\epsilon}$ to replace $g(\theta)$. $\epsilon$ is an infinitesimal and $\frac{g(\theta)}{|g(\theta)|+\epsilon} \in \{1, 0, -1\}$. Then the iterative equation is changed to formula 5.

$$
\begin{aligned}
\tilde{\theta} &= \theta_{t-1} - 2^{\eta_{t-1}} * \frac{g(\theta_{t-1})}{|g(\theta_{t-1})| + \epsilon} \\
\eta_t &= \eta_{t-1} - \beta * \frac{-g(\tilde{\theta}) * g(\theta_{t-1})}{|-g(\tilde{\theta}) * g(\theta_{t-1})| + \epsilon} \\
\theta_t &= \theta_{t-1} - 2^{\eta_t} * \frac{g(\theta_{t-1})}{|g(\theta_{t-1})| + \epsilon}
\end{aligned}
\tag{5}
$$

To solve grad loss, the key is the second change. After this change, the update stride is absolutely controlled by the learning rate and has nothing to do with the value of gradient. Then the grad loss

can be solved. Additionally, we use exponential learning rate to avoid negative value of learning rate. Otherwise parameter will move to reverse.

While in practice, the occasions of crossing the extreme point still exist under DSA. The attribution of these occasions can be divided into two types. One is that learning rate increases so fast that parameter crosses the extreme point. The other is that the learning rate decreases so slowly that DSA is not able to avoid crossing the extreme point. It affects the stability of DSA, and we will solve this problem in Section 2.2.4.

### 2.2.4 INCREASE SLOWLY AND DECREASE FAST

To reduce occasions where the parameter crosses extreme point, we use a small stride $\beta_2$ when the learning rate should increase, and a large stride $\beta_1$ in the opposite. The reason is as follows. A small stride can solve the occasion that the learning rate increases too fast and parameter crosses the extreme point. A large stride can solve the occasion that the learning rate decreases too slowly so that DSA is not able to avoid crossing the extreme point.

Thus the update equation of learning rate is changed to formula 6. According to formula 6, when learning rate should increase, $g(\tilde{\theta}) * g(\theta_{t-1})$ is positive, then $\Delta_\eta$ equals the small stride $\beta_2$. When learning rate should decrease, $g(\tilde{\theta}) * g(\theta_{t-1})$ is negative, then $-\Delta_\eta$ equals the large stride $\beta_1$.

$$\eta_t = \eta_{t-1} + \frac{\beta_1 + \beta_2}{2} * \frac{g(\tilde{\theta}) * g(\theta_{t-1})}{|-g(\tilde{\theta}) * g(\theta_{t-1})| + \epsilon} + \frac{\beta_2 - \beta_1}{2}, \beta_1 > \beta_2 \tag{6}$$

Through the above three improvement, the learning rate in DSA gets very sensitive, and DSA can finally take extremely quick and stable optimization. Then we will illustrate the method of applying DSA to a training process in Section 2.3.

### 2.3 TRAINING METHOD

In this section, we try to apply DSA to the training sessions to fit both small and large datasets. It is easy to apply DSA to solve small datasets with updating network after each epoch, which is called batch training. While for large datasets, it is necessary to divide th dataset into mini-batches and updating parameters after each mini-batch, which is called mini-batch training.

In the following, we first explain how to apply DSA to solve small datasets with batch training and then draw the method of dealing with large datasets.

We summarize the process of applying DSA to batch training in Algorithm 1 of Appendix B, which is a simple application of formula 5 and formula 6.

Now we explain how to solve large datasets with DSA. As described above, the dataset should be divided into multiple mini-batches. We can still update network after reading all the mini-batches just like a batch training. However, it will be very slow to converge and fail to capture local features of dataset. If we update parameters after each mini-batch, i.e. mini-batch training, the neural network can capture local features of dataset well and take faster optimization because of more backward propagation than batch training. However, the mini-batch training always fails to capture global features of dataset due to the excessive attention to the latest mini-batch in each optimization.

A more effective training method is a mix of them. We first train a model for several epochs with mini-batch training to capture the local features. Then we use batch training to train the model continually. In this way, both local features of datasets and global features will be captured. For the second stage, the optimizer should be able to have neural network converge fast in as few epochs as possible. Otherwise, the local features will faded out along with batch training. There is no doubt DSA is the best option for the second stage. As for the first stage, we choose Adamax as optimizer. Adamax shows a stable performance in some cases as discussed in Kingma & Ba (2014), which can be seen in our experimental result, too.

## 3 EXPERIMENT

To verify the performance of the proposed approaches, we conduct extensive experiments. In this section, we first introduce the basic settings necessary for experiments in Section 3.1. Then we will show the results and take analyses in Section 3.2. Nextly, two case studies will be conducted in Section 3.3. Finally, we will take sensitivity analysis for DSA and put the result and analyses in Appendix H.

### 3.1 EXPERIMENT SETTING

**Dataset** We applied DSA to FMP (Graham, 2014), DNN (Kunihiko & Fukushima, 1980; Lecun & Bottou, 1998) and MLP (Gardner & Dorling, 1998). On FMP, we choose the well known and widely used validation datasets MNIST (LeCun & Cortes, 2010), SVHN (svh, 2011), CIFAR10 and CIFAR100 (Krizhevsky et al., 2009). On MLP, we choose iris, wine, car and agaricus-lepiota[2] as validation data set, since they are distinguishable to different optimizers. The basic information of these datasets is shown in Appendix C.

**Baseline** According to the different optimizer's features, we choose the following 8 optimizers as the competitors, i.e., SGD (Lemaréchal, 2012), Momentum (Sutskever et al., 2013b), AdaGrad (Duchi et al., 2011), AdaDelta (Zeiler, 2012), RMSProp (Graves, 2013), Adam (Kingma & Ba, 2014), Adamax (Kingma & Ba, 2014) and AdamW (Loshchilov & Hutter, 2017). These optimizers are designed to solve different problems in the early optimizer. Among them, SGD is the early optimizer. Momentum is a speed-up optimizer. Others are learning rate adaptive optimizers.

**Metrics** We recorded the following metrics of the classifier to measure the effect of different optimizers from different perspectives, i.e., accuracy, F1-score, recall and precision of a trained neural network (Pedregosa et al., 2011). Additionally, the loss sequence along with training session is also recorded to demonstrate the convergence effect.

**Implementations** FMP is designed with reference to Graham et al. (2018); Graham & van der Maaten (2017) as the structure of FMP is visualized in Appendix D. DNN is designed with reference to Kunihiko & Fukushima (1980); Lecun & Bottou (1998); Behnke & Sven (2003) and we set channel size as a small value to simplify the model to make it more sensitive to be used to distinguish different optimizers' convergence speed. MLP sequences 5 fully connected layers. The output feature dimensions of each layer are: 32, 64, 256 and 128. The output of the first and third layers are processed using sigmoid activation. The output of the second and the fourth layers are processed by prelu activation (He et al., 2015). The features output by the neural network are processed by log_softmax activation and cross-entropy loss function. For FMP, the batchsize of MNIST, SVHN, CIFAR10, CIFAR100 is 64, 64, 128, 256 respectively, and the training epochs($T_1, T_2$) is set as (50,15), (75,15), (75,15), (50,15) respectively. For DNN, the batchsize of MNIST, SVHN, CI-FAR10, CIFAR100 is 64, 64, 64, 32 respectively, and the training epochs($T_1, T_2$) is set as (15,15). For MLP, epochs is set as 100. For DSA, $\eta_0$ is set as -12, $\beta_1$ is set as 0.6, and $\beta_2$ is set as 0.3. For SGD, the learning rate is set as 0.01. For Momentum, $\beta$ is set as 0.9, and the learning rate is set as 0.01. The remaining hyper-parameters of optimizers use the built-in values (Paszke et al., 2019). Our experiments are conducted on GTX 3090 GPU for FMP and GTX 3060Ti GPU for the others.

### 3.2 EXPERIMENTAL RESULTS

In this section, we show the results of three groups of experiments. Firstly, we apply DSA to the large convolution network FMP to show that DSA is able to optimize complex model in short epochs. Secondly, we apply DSA to a lightweight and sensitive convolution network DNN and compare it with other benchmark optimizers to show the superiority of DSA in speed and performance. Thirdly, we apply DSA to MLP to show the performance of DSA in batch training compared with baselines and the ability to process feature datasets.

To show that DSA is suitable for large datasets and complex neural network, we firstly apply DSA on FMP which is to process four image datasets and record the key metrics. We have not taken any measures that can additionally enhance the effect of the model, such as data expansion and dropout, to clearly compare the effect of optimizers. We set the training epoch 90 at most and show

---

[2]https://archive.ics.uci.edu/ml/datasets

the results in Table 1 after removing optimizers with poor performance. From the results, DSA achieves an accuracy of 99.73% when FMP is running on MNIST and an improvement of about 8 percentage on CIFAR100 compared to Adamax. Obviously, DSA can deal with large neural network and datasets in extremely short epochs and still gain extraordinary performance.

Table 1: The results on FMP. Eight different optimizers are tried. Only Adamax and Adam are competent for this task in short epochs. We have removed useless control group from this table. ACCU, F1, RC, PCS denotes accuracy, f1-score, recall and precision, respectively.

| | MNIST | | | | SVHN | | | |
| --- | --- | --- | --- | --- | --- | --- | --- | --- |
| | ACCU | F1 | RC | PCS | ACCU | F1 | RC | PCS |
| Adamax | 99.52 | 99.52 | 99.52 | 99.52 | 95.42 | 95.11 | 95.29 | 94.98 |
| DSA | **99.73** | 99.73 | 99.73 | 99.73 | **96.21** | 95.95 | 95.89 | 96.02 |
| Adam | 70.21 | 69.23 | 72.56 | 69.69 | | | | |
| | CIFAR10 | | | | CIFAR100 | | | |
| | ACCU | F1 | RC | PCS | ACCU | F1 | RC | PCS |
| Adamax | 86.91 | 86.99 | 87.17 | 86.91 | 51.13 | 51.09 | 52.89 | 51.13 |
| DSA | **88.84** | 88.84 | 88.86 | 88.84 | **59.50** | 59.63 | 60.02 | 59.50 |

To distinguish these optimizers clearly, we conduct experiments on a lightweight convolution neural network DNN referencing Kunihiko & Fukushima (1980); Lecun & Bottou (1998); Behnke & Sven (2003), where more optimizers can converge to a reasonable result in short epochs. Key part of the results is shown in Table 2, and the details are placed in Appendix E. Further, we visualize the loss trend along with epoch as Figure 8 of Appendix E. It is easy to see that DSA converges with an extremely fast speed and touches the limit of DNN far more quickly than any other optimizers.

Table 2: Result on DNN. The four metrics are accuracy, f1-score, recall and precision. In the right tabular, we stack the result of SVHN and CIFAR100 together. The first three lines are for SVHN and the others is for CIFAR100.

| MNIST | | | | | SVHN & CIFAR10 | | | | |
| --- | --- | --- | --- | --- | --- | --- | --- | --- | --- |
| SGD | 98.44 | 98.43 | 98.43 | 98.43 | AdaDelta | 60.42 | 56.39 | 57.54 | 56.69 |
| Monmentum | 95.69 | 95.65 | 95.75 | 95.63 | Adamax | 86.52 | 85.12 | 85.35 | 84.97 |
| AdaDelta | 97.35 | 97.33 | 97.37 | 97.33 | DSA | 87.35 | 86.18 | 86.25 | 86.16 |
| AdaGrad | 98.48 | 98.46 | 98.47 | 98.46 | SGD | 41.70 | 40.65 | 42.50 | 41.70 |
| Adam | 97.68 | 97.65 | 97.68 | 97.66 | Monmentum | 43.89 | 42.21 | 43.82 | 43.89 |
| AdamW | 98.42 | 98.41 | 98.40 | 98.42 | AdaGrad | 53.48 | 53.50 | 53.69 | 53.48 |
| Adamax | 98.80 | 98.79 | 98.79 | 98.79 | Adamax | 60.23 | 59.52 | 60.00 | 60.23 |
| DSA | 98.93 | 98.92 | 98.92 | 98.92 | DSA | 62.15 | 62.08 | 62.08 | 62.15 |

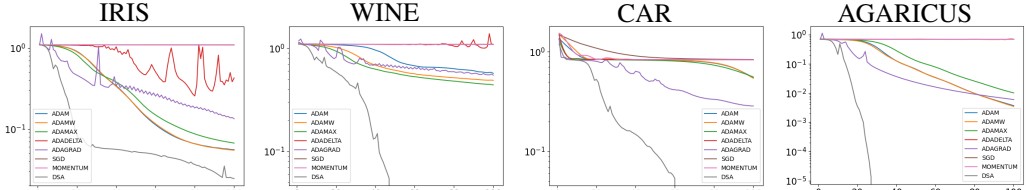

Figure 3: Loss trend on MLP.

We then conduct experiments on four small feature datasets with result in Table 7 of Appendix F to show the performance of DSA in batch training and processing feature datasets. Among the four datasets, WINE and CAR are the most discriminative datasets for DSA and other optimizers, where DSA obtains an increase of about 10 percentage in accuracy and even double performance in f1-score and recall. Loss trend is visualized in Figure 3 and DSA converges in 50 epochs, 60 epochs, 30 epochs on WINE, CAR, AGAICUS respectively, which is impossible for other optimizers. Further, we set epoch as 1,000 to see the performance of other optimizers as shown in Figure 4. We can see that other optimizers stop convergence near 0.1 on IRIS, where only DSA converge to 0. On

AGARICUS, DSA use only $\frac{1}{30}$ of other optimizer's epochs to converge to the same level, which means DSA is 30 times faster than other optimizers.

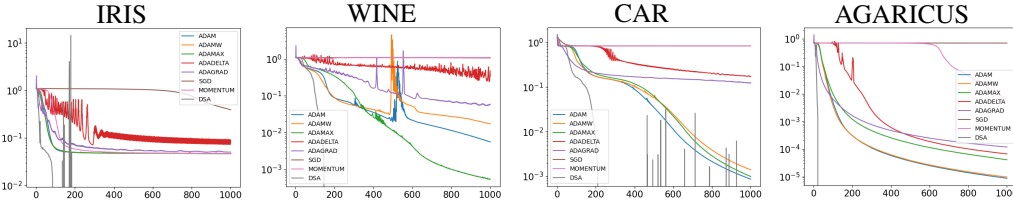

Figure 4: Loss trend on MLP with 1000 epochs.

## 3.3 CASE STUDY

In this section, we conduct two case studies. The first is a simple regression problem to calculate the sum of four numbers. The second is a minimization problem.

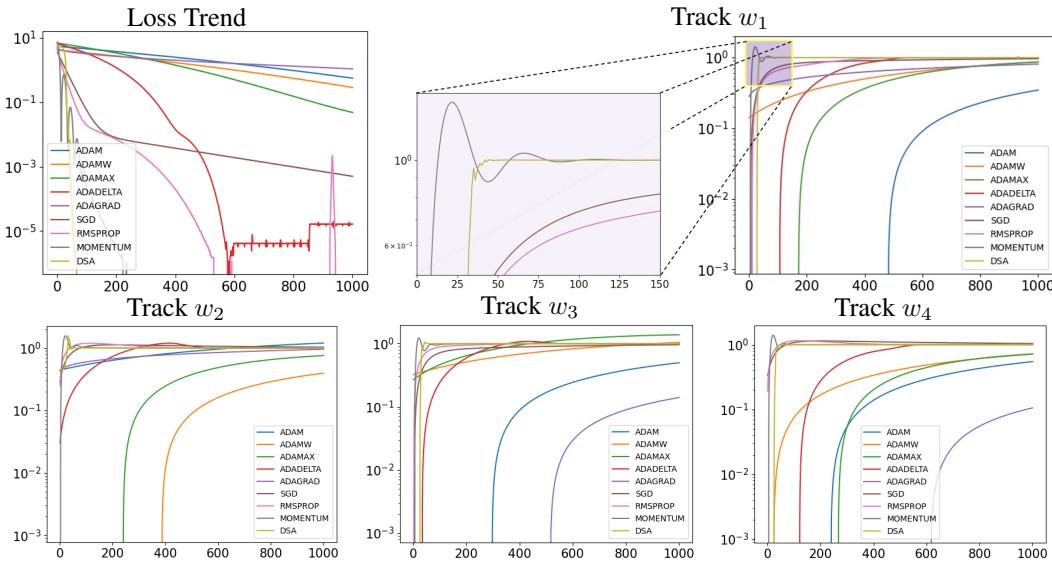

Figure 5: Result on sum of four numbers.

As the first case, we solve a simple regression problem named the sum of four numbers with target of $\min_{\boldsymbol{w}} ||\boldsymbol{w} \cdot \boldsymbol{x}^{\mathrm{T}} - y||$, where $\boldsymbol{w} = [w_1, w_2, w_3, w_4] \in \mathbb{R}^4$. Obviously, the best value of each $w_i$ is 1. We set this case to show the concrete optimization process of each optimizer and compare the convergence speed and stability among them from a more intuitive perspective. As shown in Figure 5, Momentum and DSA are the most suitable optimizers for this task. While Momentum converges quickly but unstable according to the loss trend. Therefore, DSA completes optimization more quickly. The reason can be seen in the track of $w_1$, where Momentum has a large oscillation when close to 1 and take many epochs to calm down as shown in Figure 5(Track $w_1$). While DSA solves this occasion with a quick adaptive of learning rate and quickly converge to 1.

The second case is a minimization of $\min_{w_1, w_2} a * w_1^2 + b * w_2^2$. We set this case to show the strong adaptive ability of DSA. On the one hand, the track of $(w_1, w_2)$ will always have oscillation. The smaller oscillation is, the stronger adaptive ability is. On the other hand, the change of update stride in each iteration also reflects the optimizer's adaptive ability. Our study consists of two groups, one for SGD and Momentum and the other for RMSProp, AdaGrad, Adam, Adamax and AdamW. In the first line of Figure 6, SGD has a violent oscillation along $w_2$ before calming down and moving to target slowly. Momentum also has violent oscillation along both $w_1$ and $w_2$, but it moves faster than SGD. Far more better than SGD and Momentum, DSA is able to move to the target along a straight

line without any visible oscillation. Additionally, the stride continues increasing in the beginning and decreases quickly in the end. In the second line of Figure 6(details in Appendix G), we select a hard initial position and observe the tracks. Other adaptive optimizers all experienced a big wave before calm down, especially for RMSProp and AdaGrad. Adam moves to the target more smoothly, but the speed is too slow. DSA solves all these questions, with efficiently calming down and quick movement to the target within short iterations.

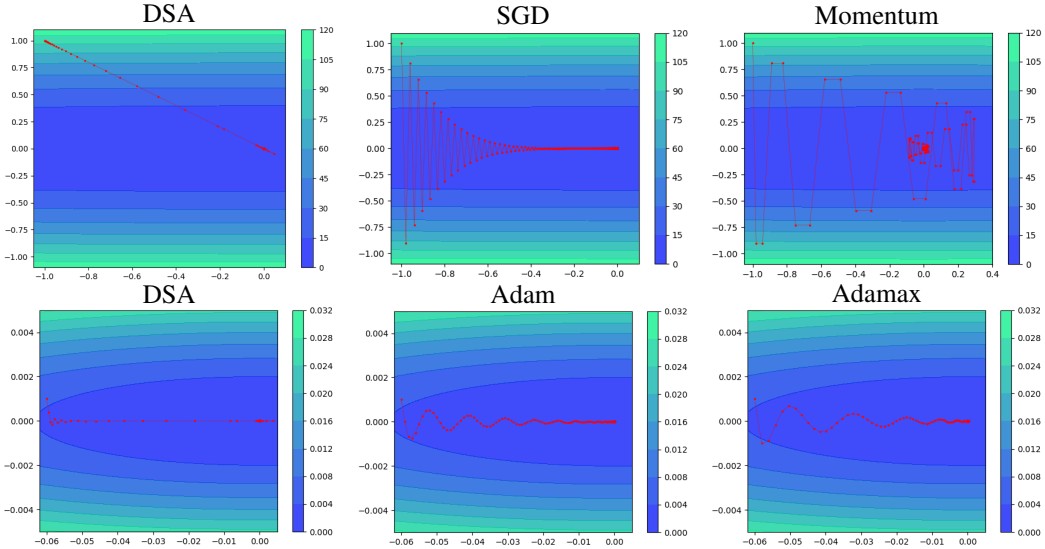

Figure 6: Track visualization. Track starts from (-0.06,0.001), $a = 1, b = 1000$.

## 4 RELATED WORK

Stochastic gradient descent (Lemaréchal, 2012) is the most commonly used optimizer. Based on a fixed learning rate, the gradient itself determines the update stride, which has serious efficiency problems and is greatly affected by grad loss. Momentum (Sutskever et al., 2013b) speeds up the training process by adding momentum information. AdaGrad (Duchi et al., 2011) adds two-norm of historical gradient on the basis of SGD so that the learning rate can be adaptive. That is, it will keep decreasing in the whole training session. RMSProp (Graves, 2013) introduces second-order momentum to solve the problem that AdaGrad is greatly affected by historical gradients. AdaDelta is also an adaptive optimizer. Although it does not have the concept of learning rate, it is analogous to Newton's method to find a more accurate stride for each update. Adam (Kingma & Ba, 2014) combines Momentum and RMSProp, so owns both of their advantages. Based on Adam, AdamW (Loshchilov & Hutter, 2017) adds a regular term to achieve a better convergence effect. Adamax extends the two norm to the infinite norm to obtain more stable and concise results. SparseAdam (Kingma & Ba, 2014) is designed to deal with sparse tensors. L-BFGS algorithm (Schmidt, 2005) is more suitable for large-scale numerical calculations, and has the characteristics of fast convergence of Newton's method. ASGD (Sutskever et al., 2013a) is asynchronous stochastic gradient descent, which is mostly used in large data parallel systems.

## 5 CONCLUSION & FUTURE WORK

In this paper, we propose the optimizer with truly self-adaptive learning rate for fast and stable convergence.Compared with existing optimizers, DSA has stronger adaptive capabilities and is competent for a variety of machine learning tasks. While this requires a reasonable initial value of the learning rate and stride size $\beta_1$ and $\beta_2$. In addition, in the later stage of training, the learning rate is still in an active state, which means pointless adaptation. Therefore, how to determine a reasonable initial value or eliminate the negative influence of the initial value and make the learning rate of DSA converge stably will be the main issue to be studied next.

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

## A  CALCULATE THE GRADIENT OF LEARNING RATE

According to the description in Section 2.2.1, we have $\tilde{L}(\eta) \triangleq L(\theta - \eta * g(\theta))$ and $\tilde{\theta} \triangleq \theta - \eta * g(\theta)$. $\eta$ is learning rate and $g(\theta)$ is gradient of $\theta$ with the goal of $\min_\theta L(\theta)$. Obviously, $\eta$'s gradient is $\frac{\mathrm{d}\tilde{L}(\eta)}{\mathrm{d}\eta}$ and we can calculate it with formula 7.

$$
\begin{aligned}
\frac{\mathrm{d}\tilde{L}(\eta)}{\mathrm{d}\eta} &= \frac{\mathrm{d}L(\theta - \eta * g(\theta))}{\mathrm{d}\eta} \\
&= \frac{\mathrm{d}L(\tilde{\theta})}{\mathrm{d}\eta} \\
&= \sum_{\tilde{\theta}} \frac{\partial L(\tilde{\theta})}{\partial \tilde{\theta}} * \frac{\partial \tilde{\theta}}{\partial \eta} \\
&= \sum_{\theta,\tilde{\theta}} \frac{\partial L(\tilde{\theta})}{\partial \tilde{\theta}} * \frac{\partial \theta - \eta * g(\theta)}{\partial \eta} \cdots \tilde{\theta} \triangleq \theta - \eta * g(\theta) \\
&= \sum_{\theta,\tilde{\theta}} g(\tilde{\theta}) * \left( \frac{\partial(\theta - \eta * g(\theta))}{\partial \eta} \right) \\
&= \sum_{\theta,\tilde{\theta}} (-g(\tilde{\theta}) * g(\theta))
\end{aligned}
\tag{7}
$$

## B  PSEUDO CODE OF DSA WHEN APPLIED TO BATCH TRAINING

**Algorithm 1** In Line 2-3, we take once forward and backward propagation to calculate the gradient for $\tilde{\theta}$ which is used for calculating the learning rate's gradient. In Line 4-5, we compute the learning rate's gradient and update the learning rate according to its gradient. In Line 6, we finally update network's parameter based on updated the learning rate and gradient calculated in Line 2.

---

**Input:** neural network model $f(\theta_0)$, data set $\boldsymbol{X}, \boldsymbol{y}$, initial learning rate $\eta_0$, learning rate update step size $\beta_1$ and $\beta_2$, the number of iterations $T$, infinitesimal $\epsilon$
**Output:** trained neural network model $f(\theta_T)$
1: **for** $t : 1 \mapsto T$ **do**
2:  $\quad \hat{\boldsymbol{y}} \leftarrow f(\boldsymbol{X}|\theta_{t-1}), g(\theta_{t-1}) \leftarrow \nabla_{\theta_{t-1}}$  $\qquad\qquad\qquad$ ▷ Calculate gradients of $\theta_{t-1}$
3:  $\quad \tilde{\theta} \leftarrow \theta_{t-1} - 2^{\eta_{t-1}} * \frac{g(\theta_{t-1})}{|g(\theta_{t-1})|+\epsilon}$  $\qquad\qquad$ ▷ Calculate $\tilde{\theta}$ according to formula 5
4:  $\quad \hat{\boldsymbol{y}} \leftarrow f(\boldsymbol{X}|\tilde{\theta}), g(\tilde{\theta}) \leftarrow \nabla_{\tilde{\theta}}$  $\qquad\qquad\qquad\qquad$ ▷ Calculate gradients of $\tilde{\theta}$
5:  $\quad \eta_t \leftarrow \eta_{t-1} + \frac{\beta_1+\beta_2}{2} * \frac{g(\tilde{\theta})*g(\theta_{t-1})}{|-g(\tilde{\theta})*g(\theta_{t-1})|+\epsilon} + \frac{\beta_2-\beta_1}{2}$  $\quad$ ▷ Update learning rate $\eta$ according to formula 6
6:  $\quad \theta_t \leftarrow \theta_{t-1} - 2^{\eta_t} * \frac{g(\theta_{t-1})}{|g(\theta_{t-1})|+\epsilon}$  $\qquad$ ▷ Update network's parameters according to formula 5
7: **end for**

---

## C   META INFORMATION OF DATASETS

Table 3: Dataset information

| Dataset | #Train/ #Test | #Attributes | #Class |
|---|---|---|---|
| MNIST | 60,000 /10,000 | 1*28*28 | 10 |
| SVHN | 73,257 /26,032 | 3*32*32 | 10 |
| CIFAR10 | 50,000 /10,000 | 3*32*32 | 10 |
| CIFAR100 | 50,000 /10,000 | 3*32*32 | 100 |
| IRIS | 120/ 30 | 4 | 3 |
| WINE | 142/ 36 | 13 | 3 |
| CAR | 1,382/ 346 | 6 | 4 |
| AGARICUS | 6,499 / 1,625 | 116 | 2 |

## D   STRUCTURE OF FMP

We visualize the structure of FMP in Figure 7 supposing the input is a image of $28 \times 28$. FMP is composed of 6 convolution block and 1 linear block. Each convolution block ends up with a fractional maxpool. Each output of a convolution layer is processed by prelu activation.

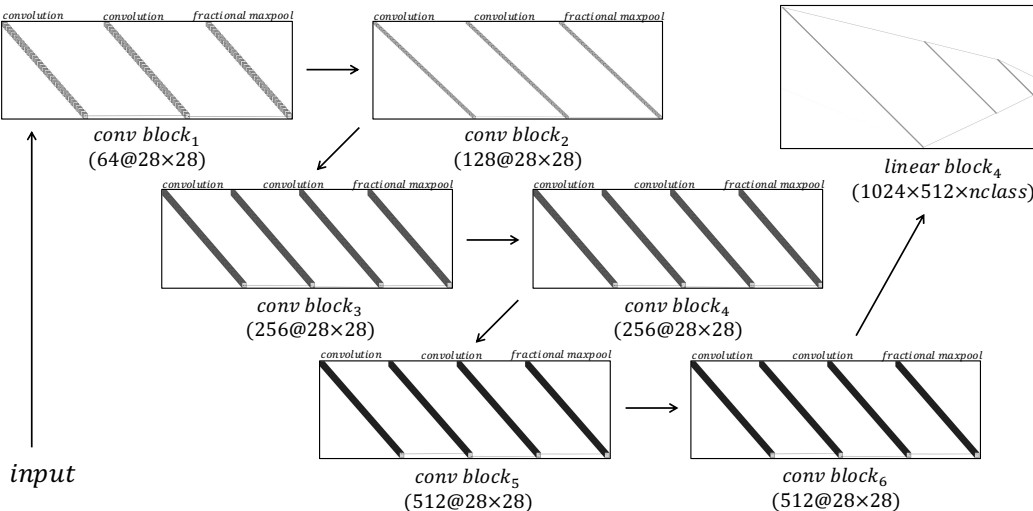

Figure 7: Structure of FMP.

# E   RESULTS OF LARGE DATASETS IN DETAIL

Table 4: Dnn MNIST result detail

| | MNIST | | | |
| --- | --- | --- | --- | --- |
| | ACCU | F1-SCORE | RECALL | PRECISION |
| SGD | 98.44 | 98.43 | 98.43 | 98.43 |
| Monmentum | 95.69 | 95.65 | 95.75 | 95.63 |
| RMSProp | 11.35 | 2.04 | 1.14 | 10 |
| Adadelta | 97.35 | 97.33 | 97.37 | 97.33 |
| Adagrad | 98.48 | 98.46 | 98.47 | 98.46 |
| ADAM | 97.68 | 97.65 | 97.68 | 97.66 |
| ADAMW | 98.42 | 98.41 | 98.4 | 98.42 |
| ADAMax | 99.02 | 99.02 | 99.02 | 99.02 |
| DSA | 98.93 | 98.92 | 98.92 | 98.92 |

Table 5: Dnn SVHN result detail

| | SVHN | | | |
| --- | --- | --- | --- | --- |
| | ACCU | F1-SCORE | RECALL | PRECISION |
| SGD | 19.59 | 3.28 | 1.96 | 10 |
| Monmentum | 19.59 | 3.28 | 1.96 | 10 |
| RMSProp | 19.59 | 3.28 | 1.96 | 10 |
| Adadelta | 60.42 | 56.39 | 57.54 | 56.69 |
| Adagrad | 19.59 | 3.28 | 1.96 | 10 |
| ADAM | 19.59 | 3.28 | 1.96 | 10 |
| ADAMW | 19.59 | 3.28 | 1.96 | 10 |
| ADAMax | 86.52 | 85.12 | 85.35 | 84.97 |
| DSA | 87.35 | 86.18 | 86.25 | 86.16 |

Table 6: Dnn cifar10 result detail

| | CIFAR10 | | | |
| --- | --- | --- | --- | --- |
| | ACCU | F1-SCORE | RECALL | PRECISION |
| SGD | 41.7 | 40.65 | 42.5 | 41.7 |
| Monmentum | 43.89 | 42.21 | 43.82 | 43.89 |
| RMSProp | 10 | 1.82 | 1 | 10 |
| Adadelta | 40.52 | 40.55 | 43.11 | 40.52 |
| Adagrad | 53.48 | 53.5 | 53.69 | 53.48 |
| ADAM | 10 | 1.82 | 1 | 10 |
| ADAMW | 10 | 1.82 | 1 | 10 |
| ADAMax | 60.23 | 59.52 | 60 | 60.23 |
| DSA | 62.15 | 62.08 | 62.08 | 62.15 |

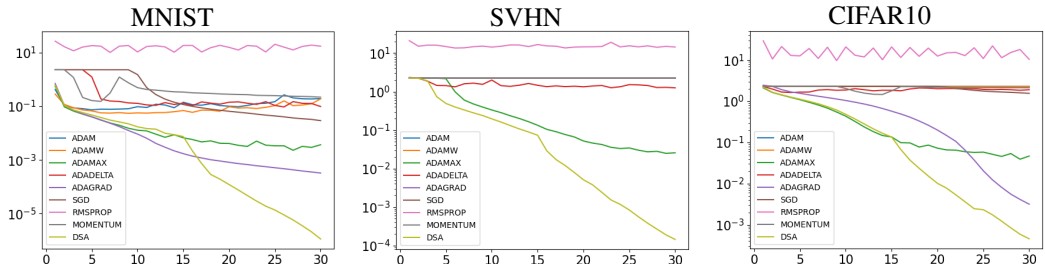

Figure 8: Loss trend on DNN with different optimizers.

# F    RESULTS OF SMALL DATASETS IN DETAIL

Table 7: Result on MLP with four feature datasets.

| | IRIS | | | | WINE | | | |
|---|---|---|---|---|---|---|---|---|
| | ACCU | F1 | RC | PCS | ACCU | F1 | RC | PCS |
| SGD | 20.00 | 11.11 | 06.67 | 33.33 | 77.78 | 56.91 | 52.70 | 62.20 |
| Monmentum | 20.00 | 11.11 | 06.67 | 33.33 | 66.67 | 48.53 | 44.44 | 53.57 |
| RMSProp | 43.33 | 20.16 | 14.44 | 33.33 | 44.44 | 20.51 | 14.81 | 33.33 |
| Adadelta | 100.0 | 100.0 | 100.0 | 100.0 | 80.56 | 59.02 | 55.19 | 64.29 |
| Adagrad | 100.0 | 100.0 | 100.0 | 100.0 | 80.56 | 59.02 | 55.19 | 64.29 |
| ADAM | 100.0 | 100.0 | 100.0 | 100.0 | 80.56 | 59.45 | 56.52 | 64.29 |
| ADAMW | 100.0 | 100.0 | 100.0 | 100.0 | 80.56 | 59.02 | 55.19 | 64.29 |
| ADAMax | 100.0 | 100.0 | 100.0 | 100.0 | 83.33 | 79.10 | 78.89 | 80.26 |
| DSA | 100.0 | 100.0 | 100.0 | 100.0 | 100.0 | 100.0 | 100.0 | 100.0 |

| | CAR | | | | AGARICUS | | | |
|---|---|---|---|---|---|---|---|---|
| | ACCU | F1 | RC | PCS | ACCU | F1 | RC | PCS |
| SGD | 69.36 | 20.48 | 17.34 | 25.00 | 52.43 | 34.40 | 26.22 | 50.00 |
| Monmentum | 69.36 | 20.48 | 17.34 | 25.00 | 52.43 | 34.40 | 26.22 | 50.00 |
| RMSProp | 69.36 | 20.48 | 17.34 | 25.00 | 52.43 | 34.40 | 26.22 | 50.00 |
| Adadelta | 69.36 | 20.48 | 17.34 | 25.00 | 90.34 | 90.23 | 90.98 | 90.05 |
| Adagrad | 88.73 | 59.47 | 54.83 | 66.86 | 99.94 | 99.94 | 99.94 | 99.94 |
| ADAM | 77.46 | 33.05 | 33.35 | 33.86 | 99.94 | 99.94 | 99.94 | 99.94 |
| ADAMW | 77.46 | 33.05 | 33.35 | 33.86 | 99.94 | 99.94 | 99.94 | 99.94 |
| ADAMax | 74.28 | 29.12 | 30.23 | 30.38 | 99.75 | 99.75 | 99.76 | 99.75 |
| DSA | 98.55 | 96.31 | 97.68 | 95.08 | 100.0 | 100.0 | 100.0 | 100.0 |

## G  RESULTS OF TRACK ADAPTIVE OPTIMIZERS IN DETAIL

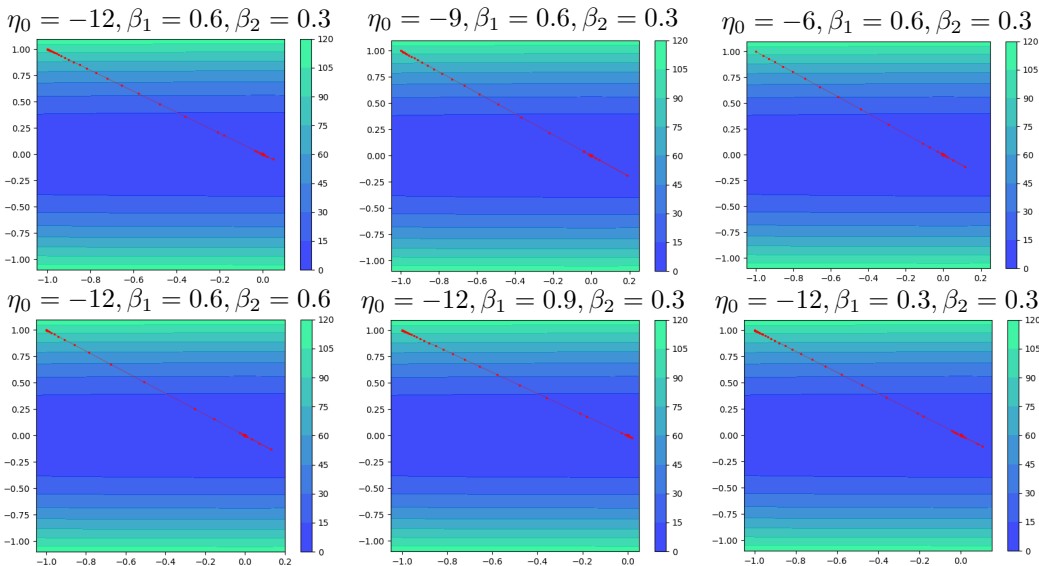

Figure 9: Start from (-0.06,0.001), $a = 1, b = 1000$

## H  SENSITIVITY ANALYSIS

In this section, we analyse the effect of different hyper-parameters of DSA. There are 3 hyper-parameters in DSA, i.e., the initial value of learning rate $\eta_0$, updating stride $\beta_1$ and $\beta_2$. The larger $\eta_0$ and $\beta_2$ is, the faster DSA moves in the beginning. The larger $\beta_1$ is, the faster DSA converges in the end. A group of DSA's tracks with different hyper-parameters are contained in Figure 10. When $\eta_0$ or $\beta_2$ increases, the convergence will speed up because of larger stride. When $\beta_1$ increases, the convergence will speed up because of the stronger adaptive capacity.

Figure 10: Start from (-1,1), $a = 1, b = 95$

