# OpenReview forum: "Differentiable Self-Adaptive Learning Rate"
_ICLR.cc/2022/Conference — ICLR 2022 Submitted_

### Official Review · Reviewer_bLyu · 2021-10-21

**Correctness:** 3
**Technical Novelty And Significance:** 1
**Empirical Novelty And Significance:** 1
**Recommendation:** 1
**Confidence:** 5

**Main Review:**

The presented method is already known, the authors do not cite the preceding works in that line of research, and pretend that "[they] are the first to make learning rate in neural network differentiable with the goal of minimizing loss function".

Some non-cited preceding papers:
 * Seminal paper: *Learning to learn by gradient descent by gradient descent*, Andrychowicz 2016.
 * Paper with explicit optimization of $\eta$ by gradient descent in the context of feeforward neural networks: *Meta-SGD: Learning to Learn Quickly for Few-Shot Learning*, Li 2018.
 * Early preprint in the same in the same line of work: *Speed learning on the fly*, Massé 2016.

Besides, the notation lacks rigour:
 * $\tilde{L}(\eta)$ is not truly a function of the learning rate $\eta$, it also depends on the parameters $\theta$.
 * The use of $\tilde{\theta}$ for the SGD update of $\theta$ is a bit clumsy: using $\theta_t$ and $\theta_{t + 1}$ would have been clearer. Besides, the use of $\tilde{\theta}$ without index $t$ to denote the training step shadows some freedom that we have when using the algorithm: we may want to train $\eta$ each $T$ steps (with arbitrary $T > 0$) instead of each $1$ step.

**Summary Of The Paper:**

The authors propose to tune the learning rate of a SGD, by training it by gradient descent.

**Summary Of The Review:**

This work is practically contained in existing papers, which are ignored by the authors. Strong reject.

---

> ### Author Response · Authors · 2021-11-09
> **Sorry for our serious mistake**
>
> Thanks for pointing out our serious mistake. You have proposed the same fake in our method with the first reviewer. As described in paper, we have make learning rate specific for each network's parameter, which is the core to achieve good perforance. However, we consider making the learning rate differentiable as our greatest contribution by mistake.
>
> As for $\tilde{L}(\eta)$, it's truly a function of $\eta$, cause we have got the gradient of $\theta$ here and want to know how to use gradient to make the global loss decrease faster. So it can be viewed as a minimization of loss with respect to $\eta$.

---

### Official Review · Reviewer_piNT · 2021-11-01

**Correctness:** 4
**Technical Novelty And Significance:** 3
**Empirical Novelty And Significance:** 3
**Recommendation:** 8
**Confidence:** 5

**Main Review:**

Authors propose an optimizer called DSA, which can increase or decrease very sensitively and significantly with the help of the learning rate’s gradient, which overcomes the disadvantages of the existing algorithms e.g., AdaGrad, RMSProp, and Adam. The proposed DSA is the first optimizer that has a clear instruction of directing the loss function, which applies to a wide range of DNN models as well as two classical machine learning problems. The paper has presented extensive experiments and proofed the values of DSAN as what I believe so far.

**Summary Of The Paper:**

The authors propose a novel optimizer that makes the learning rate of a deep neural network to be differentiable in a self-adaptive manner, which overcomes the problems of existing problems DNN optimizers i.e., insensitive and unstable.

**Summary Of The Review:**

The paper is ready for acceptance.

---

> ### Author Response · Authors · 2021-11-09
> **Thanks for your approval**
>
> Thanks for your approval greatly.

---

### Official Review · Reviewer_thhp · 2021-11-04

**Correctness:** 3
**Technical Novelty And Significance:** 2
**Empirical Novelty And Significance:** 2
**Recommendation:** 3
**Confidence:** 5

**Main Review:**

This paper is well organized with an intuitive explanation and extensive experiments, and it can be considered as a good empirical study on investigating adaptive learning rates for solving machine learning problems. However, there are a few weaknesses of the paper, and I am listing them (with questions that could improve reviewers' understanding) here.

1: First of all, the DSA method is in fact minimizing the loss function given current iterate $\theta$ and gradient $g$. Thus, if the modified loss function can be solved exactly, it would produce the optimal $\eta$ on the direction with current iterate and gradient. However, how could the author justify that it is better than a constant (or indeed any random) learning rate? Can you justify that in theory?

2. Secondly, the novelty is weak as more literature review in optimization field would give rich content related to the DSA method. The classical way of deriving adaptive learning rate (mostly for the deterministic problem, but also stochastic problem in more recent literature) is line search. Line search finds appropriate learning rate along the same direction that DSA does. In addition, implicit SGD[1] was developed with implicit schema to derive a learning rate. Also, more recent work of deriving adaptive learning rate was proposed for variance reduced method [2].

3. Indeed, the modified loss function is a one-dimension optimization sub-problem. The way that DSA deals with the sub-problem is to either propose a gradient descent or use hyper-parameter to perform one-step update. Could you solve it approximately by using the newton method? In [2], a predefined sub-problem was solved approximately by newton method.

4. As authors define the terms in the paper, such as "stable", "sensitive", "grad loss" etc, this paper tries to explain the issues that was caused by non-adaptive method. However, the proposed DSA method is not solving the challenging optimization problem in the field. Could the weakness of convergence (as authors mention) for other optimizers be analyzed more? For example, optimizer often struggles to escape the saddle point, Adam with constant step-size only converges to neighborhood of local minimum, etc. I strongly feel that more in-depth understanding of issues with classical optimizers is needed for the proposed method to address the concerns in the field.

5. The empirical study is extensive. However, a few confusion that I am having are:

5.1 How DSA and other algorithms were configured? It seems like DSA also adopts batch training.

5.2 Does DSA incur additional cost? Could it be more fair to compare algorithms with respect to computational cost instead of epoch?

5.3 The plots for oscillation are specific for some problems, and would it cover general case?

5.4 Other algorithm requires fine-tuned learning rate, would it be fair to compare with fine-tuned version?


References:
[1] see https://sites.google.com/view/panos-toulis/implicit-sgd
[2] see https://arxiv.org/abs/2102.09700


**Summary Of The Paper:**

This paper presents a method of adaptive learning rate in training deep neural networks and solving machine learning problems. The proposed method derives the adaptive learning rate by minimizing the loss function on the direction of $\\theta - \eta g$, with $\eta$ being treated as a parameter.

**Summary Of The Review:**

In summary, I'd love to see improvement in this paper. Based on my understanding of the field, I strongly recommend authors analyze the pro and cons of other algorithms and investigate in-depth the issues of not having a truly adaptive learning rate. with adaptive learning rate in developing optimization methods, it is often necessary to include theoretical analysis. And, theoretical analysis and empirical study can work together to understand the proposed method better.

---

> ### Author Response · Authors · 2021-11-09
> **Thanks for your valuable advice**
>
> 1. We can see it in Figure~1. If we use a const learning rate, the loss will be $L(\theta_0 - \eta_0 * g_0)$. If we update learning rate with its gradient, the loss will be $L(\theta_0 - \eta_1* g_0)$. Obviously, $L(\theta_0 - \eta_1* g_0) < L(\theta_0 - \eta_0* g_0)$ with the target of minimizing $L(\theta - \eta * g(\theta))$ wrt $\eta$.
>
> 2. To tell the truth, we have make learning rate specific for each network's parameter, which is the core technology to achieve good perforance. However, we consider making the learning rate differentiable as our greatest contribution by mistake. Other two reviewers point out the same fakes. We will redefine the novelty of our work.
>
> 3. Thanks for your idea, we will try newton method in the later(Recently, I am stuck in dealing with my final exam of this term).
>
> 4. Thanks for this advice greatly, we will take more clear expression about how DSALR is sensitive and in-deep analysis about other model.
>
> 5.1 DSA is good at capture global features, so it's more suitable to take batch-train with DSA. We train the network with Adamax for several epochs with mini-batches, and then continuely train it with DSA with batch-train. The former is to capture the details in dataset and DSA capture the global features. And the second stage always can finish in very few epochs.
>
> 5.2 There are two forward propagation and one backward propagation. We will consider your advice and take a more fair standard.
>
> 5.3 The plots for oscillation reveals the reason why adam and sgd will have oscillation and why DSA can solve those occasions. So it's general.
>
> 5.4 In the future work, we will add lr scheduler for baseline and other training techniques to make reasonable baseline's result.
>
> Thanks for your serious review greatly again.

---

> > ### Comment · Reviewer_thhp · 2021-11-21
> > **Response to Authors**
> >
> > $\textbf{Thank you very much for your response to my comments! }$
> >
> > I'd like to follow up on a few points.
> >
> > It is not necessary that, by deriving the gradient wrt $\eta$ and then update $\eta_0$ with a multiple or fractional of the gradient, $L(\theta_0 - \eta_1 g_0) \leq L(\theta_0 - \eta_0 g_0)$. To make the inequality hold, you need to perform a search of $\eta$, and that is essentially making the problem a one-dimensional optimization problem. However, there is certainly more in-depth analysis involved. For example, even if you can solve the aforementioned one-dimension optimization problem exactly, i.e. to derive the optimal $\eta$, for every iteration, would it be necessarily the case that the resulting reductions in the loss function be better than a constant $\eta$? It might not even be the case for a deterministic setting. It requires more in-depth analysis to answer the question. Usually, it requires theoretical analysis as well as good empirical evidence.
> >
> > For the empirical study, it is not clear to me the setting of the DSA. It is probably fine to use Adamax for warm-up, but running DSA in a deterministic setting and comparing it with other algorithms in the stochastic setting is not fair. The most important reason is that they are essentially solving different problems per iteration. Also, if DSA is solving deterministic problems, then it should be compared with gradient descent with, for example, line search, or other methods that can determine good step-size.
> >
> > Overall, I'd recommend authors strengthen the paper with a more in-depth analysis. The adaptive step-size is certainly of interest in the field.

---

> > > ### Author Response · Authors · 2021-11-22
> > > **Thanks for you replies!**
> > >
> > > I would like to take a simple explaination about the mentioned **one-dimensional optimization problem**.
> > >
> > > To tell the truth, the method is not something like grid search. As described in Formula~6, we can use history gradients of network's parameters to update the learning rate directly without a search of $\eta$.
> > >
> > > So one-dimensional optimization problem doesn't exist in our method finally.

---

> > > > ### Comment · Reviewer_thhp · 2021-11-29
> > > > **Thank you, Authors!**
> > > >
> > > > Thank you for your response and explanation! After carefully reviewing your rebuttal and other reviewers' comments, I decided to keep my score. Your response was really helpful for me to understand the idea of the paper, but my initial thoughts still hold. I'd recommend authors make revisions to the paper. As I said, the direction is certainly of interest in the field.
> > > >
> > > > Thank you!

---

### Official Review · Reviewer_57Xy · 2021-11-09

**Correctness:** 2
**Technical Novelty And Significance:** 1
**Empirical Novelty And Significance:** 1
**Recommendation:** 3
**Confidence:** 5

**Main Review:**

The paper has serious shortcomings.

- The authors claim to introduce a novel method in gradient-based hyperparameter optimization (adjusting the learning rate by using the derivative of an objective with respect to the learning rate) when this has been done previously many times including by some highly cited papers such as

Maclaurin, D., Duvenaud, D. and Adams, R. Gradient-based hyperparameter optimization through reversible learning. ICML 2015. https://arxiv.org/abs/1502.03492

Baydin, A.G., Cornish, R., Rubio, D.M., Schmidt, M. and Wood, F. Online learning rate adaptation with hypergradient descent. ICLR 2018. https://arxiv.org/abs/1703.04782

and a series of more recent papers covering similar approaches. The first paper deriving this method is actually from the 1990s (Almeida et al., 1998) and this is explained in the ICLR 2018 paper listed above.

- The method they claim to introduce and its derivation has the same analytical form as the method in the paper from ICLR 2018 listed above. So they are repeating the derivation of a method previously published at least twice without having any review or mention of related work which is quite easy to find.

- The submitted paper does not have any coverage whatsoever of related work on gradient-based hyperparameter optimization, and the related works section and introduction cover only very basic level adaptive learning rate algorithms such as AdaGrad or RMSProp. A good pointer for the authors to discover and survey related work would be to start with this paper by Bengio (2000):

Y. Bengio. Gradient-based optimization of hyperparameters. Neural Computation, 12(8):1889–1900, 2000. doi: 10.1162/089976600300015187.

and follow the trail of papers citing this paper up to our current decade.

In addition to these serious shortcomings, the paper is written in a way that makes vague and unsubstantiated statements such as:

- “we … realize an optimizer with truly self-adaptive learning rate”
- “our optimizer achieves fast and high qualified convergence in extremely short epochs”
- “Adam usually cannot converge well in the late stages of the training”
- “In a word, Adam is not sensitive enough”
- “None of the current optimizers are stable and sensitive enough to ensure a fast and high qualified convergence”

and many others throughout the paper.


**Summary Of The Paper:**

The paper concerns an adaptive learning rate algorithm based on the derivative of the objective with respect to the learning rate. He authors introduce the technique that they call “differentiable self-adaptive learning rate” (DSA) and present experimental results comparing their technique with conventional optimizers including Adam, AdaGrad, SGD with momentum in tasks including classification with MNIST, CIFAR10, CIFAR100, SVHN datasets.

**Summary Of The Review:**

The paper makes the incorrect claim that the presented technique is novel and it crucially fails at finding and citing any instances of related work which are very easy to find with simple search terms like “gradient-based optimization of hyperparameters”.

I believe that this can be an honest case of an independent rediscovery of the technique by the authors combined with a lack of experience in finding related work and writing the paper in a more grounded way (e.g., avoiding vague statements). I believe that there is value in the experiments presented by the authors and I would like to encourage the authors to keep working in this problem space and improving their manuscript to get it eventually published after addressing the crucial shortcomings and finding a new angle to present their contributions.

---

> ### Author Response · Authors · 2021-11-09
> **Thanks for pointing out our mistake**
>
> Firstly, thanks for pointing out our mistake. This mistake is caused by our one-sided understanding of historical work.
> Secondly, we will continue to improve our expression in the paper.
> To tell the truth, we have make learning rate specific for each network's parameter, which is the core technology to achieve good perforance. However, we consider making the learning rate differentiable as our greatest contribution by mistake. Thanks again, we will continue to improve it.

---

### Decision · Program_Chairs · 2022-01-20

**Decision:**

Reject

**Comment:**

The paper deals with the problem of adjusting the learning rate during gradient descent optimisation. Unfortunately the proposed approach is very similar to methods already presented in the literature and no significant contribution can be recognised. During the rebuttal, the author(s) have acknowledged their ignorance about the relevant literature and provided some further clarifications that did not turn into a revision of the reviewers’ initial assessment of the work.